# *Pseudomonas aeruginosa*: Recent Advances in Vaccine Development

**DOI:** 10.3390/vaccines10071100

**Published:** 2022-07-08

**Authors:** Matthew Killough, Aoife Maria Rodgers, Rebecca Jo Ingram

**Affiliations:** 1Wellcome-Wolfson Institute for Experimental Medicine, Queen’s University Belfast, Belfast BT7 1NN, UK; mkillough03@qub.ac.uk; 2Department of Biology, The Kathleen Lonsdale Institute for Human Health Research, Maynooth University, R51 A021 Maynooth, Ireland; aoife.rodgers@mu.ie

**Keywords:** *Pseudomonas aeruginosa*, vaccines, cystic fibrosis, antibiotic resistance, mucosal immunity

## Abstract

*Pseudomonas aeruginosa* is an important opportunistic human pathogen. Using its arsenal of virulence factors and its intrinsic ability to adapt to new environments, *P. aeruginosa* causes a range of complicated acute and chronic infections in immunocompromised individuals. Of particular importance are burn wound infections, ventilator-associated pneumonia, and chronic infections in people with cystic fibrosis. Antibiotic resistance has rendered many of these infections challenging to treat and novel therapeutic strategies are limited. Multiple clinical studies using well-characterised virulence factors as vaccine antigens over the last 50 years have fallen short, resulting in no effective vaccination being available for clinical use. Nonetheless, progress has been made in preclinical research, namely, in the realms of antigen discovery, adjuvant use, and novel delivery systems. Herein, we briefly review the scope of *P. aeruginosa* clinical infections and its major important virulence factors.

## 1. An Introduction to *Pseudomonas aeruginosa*

The World Health Organisation (WHO) has prioritised *Pseudomonas aeruginosa* as one of the top three critical pathogens requiring urgent research into new treatments [1]. Alongside *Enterococcus faecium, Staphylococcus aureus, Klebsiella pneumoniae, Acinetobacter baumannii,* and *Enterobacter* spp., *P. aeruginosa* is one of the multidrug resistant (MDR) ESKAPE pathogens [2]. These pathogens contribute to significant morbidity, mortality, and cost for healthcare infrastructures across the globe [3]. A recent systematic review and meta-analysis by Lansbury et al. [4] identified *P. aeruginosa* as the second most common bacterial co-infection isolated from patients with COVID-19. This causes notable exacerbations of disease and an increase in the complexity of clinical management.

*P. aeruginosa* is a virulent bacterium, with its cell surface structures, secreted compounds and biofilm formation being the major contributors to this bacteria’s pathogenicity. The combination of virulence and antibiotic resistance allows *P. aeruginosa* to optimally cause complex infections in vulnerable immunocompromised patients, which are challenging to treat. With the increasingly frequent acquisition of MDR strains, we are approaching a time where there are no remaining therapies to treat left *P. aeruginosa* infection. As such, novel preventative and therapeutic strategies are warranted. Vaccines are a promising alternative to antibiotics to help prevent *P. aeruginosa* infections in susceptible individuals. Unfortunately, despite extensive research efforts, there are currently no licensed vaccines available. In this review, we provide an account of vaccine clinical trials against *P. aeruginosa* and discuss recent advances and developments in pre-clinical research which aim to develop an efficacious vaccine. To provide insights into the complex nature of this pathogen, and thereby the challenges surrounding development of an efficacious vaccine, we begin by providing an overview of the key virulence factors important in the pathogenesis of *P. aeruginosa* infections and subsequent infections caused by this pathogen.

## 2. Virulence Factors

The pathogenic profile of *P. aeruginosa* stems from a variable arsenal of virulence factors which together cause infection throughout the body at a range of sites. While many such factors are highly immunogenic, they facilitate the bacteria’s ability to evade and counteract the host immune response. A comprehensive overview of such factors is beyond the scope of this review, and therefore we refer the reader to recently published reviews for more in-depth information on this area [5,6]. Herein, we provide an overview of the main virulence factors of pivotal importance to the pathogenesis of *P. aeruginosa*, as shown in Figure 1.

### 2.1. Biofilm Formation

Biofilms are a hugely complex architecture of cells and a matrix of extracellular secreted compounds. Cells within biofilms adhere to a surface via flagella and type IV pili and multiply to form microcolonies [7]. Further division and matrix production provide structural stability and allow the biofilm to grow and expand [8]. The matrix consists of three exopolysaccharides (Psl, Pel and alginate) [9], which stabilise the overall structure while additionally functioning to impair bacterial clearance and a variety of host immune responses [10]. Extracellular DNA is released via cell lysis and the formation of neutrophil extracellular traps (NETs), which provide a range of functions [9]. The resulting biofilm is extremely resistant to immune clearance, environmental changes, and, importantly, antibiotics.

### 2.2. Antimicrobial Resistance

Antimicrobial resistance (AMR) in *P. aeruginosa* not only stems from the formation of biofilms, there are a variety of other important factors. These can be categorised into intrinsic, acquired, and adaptive mechanisms, as outlined in Table 1.

LPS reduces the outer membrane’s permeability to antibiotics [11]. Additionally, more than 95% of outer-membrane proteins (OMPs) are in a closed confirmation [12]. Efflux pumps and antibiotic-inactivating enzymes together inactivate/pump out many major families of antimicrobials effective against *P. aeruginosa*, such as aminoglycosides, β-lactams, and quinolones [12]. Gene mutations can significantly increase expression of these intrinsic mechanisms in response to environmental stressors [13], which are often numerous during the host immune response. Horizontal transfer of acquired resistance genes can drastically amplify resistance across many *P. aeruginosa* strains [14]. *P. aeruginosa* has the potential to form polymicrobial biofilms and infections, which are facilitated through horizontal gene transfer [15]. These mechanisms of antibiotic resistance unfortunately contribute to a huge burden of MDR infections, which are difficult to eradicate [16].

## 3. *P. aeruginosa* Infections

*P. aeruginosa* is associated with a diverse spectrum of clinical disease of varying duration and severity. *P. aeruginosa* can infect and colonise almost all body systems in humans, as shown in Figure 2. Diseases range from mild local infections to life-threatening burn wound infections, bacteraemia, and pneumonia, among others. As an opportunistic pathogen, it can cause disease in vulnerable immunocompromised patients, and is a particular burden in intensive care units (ICUs) [16]. The potential target population which would benefit from an available vaccine is wide-reaching and diverse, encompassing a vast range of distinct groups.

The skin plays a vital role in the innate immune system as an anatomical barrier [17]. Moderate and severe burns predispose the affected individual to severe *P. aeruginosa* infection, which often requires specialist management [18]. Burn patients are at risk of developing profound systemic bacteraemia and sepsis, with an estimated mortality of 37.5% [19]. A recent study concluded that isolation of *P. aeruginosa* from any previous culture from any location increases the risk of developing bacteraemia with this pathogen [19]. In the UK, between 2009 and 2018, the prevalence of *P. aeruginosa*-associated bacteraemia rose by 10.7% to 7.8 per 100,000 population [20]. In military personnel, *P. aeruginosa* is the second most common bacteria isolated from combat-associated wounds [21].

*P. aeruginosa* is a notorious pathogen in nosocomial infections, particularly ventilator-associated pneumonia [22], central line-associated bacteraemia [23], and complicated catheter-associated urinary tract infections [24]. Administration of antibiotics and/or prior hospital/ICU stay increases the risk of acquiring MDR *P. aeruginosa* [16]. This prior colonisation increases the risk of developing ventilator-associated pneumonia by up to eight times [22], and is associated with substantial morbidity and mortality [25]. Mucoid biofilm formation permits *P. aeruginosa* to colonise medical equipment, including ventilators, catheters, and bronchoscopes [26,27]. Interestingly, a recent review outlined the burden of highly adapted biofilms within healthcare-associated plumbing devices [28]. These may act as a reservoir for infection and is an excellent demonstration of the bacteria’s ability to survive and adapt. *P. aeruginosa* is not only a significant issue in hospitals, it is a common causative organism in community-acquired pneumonia [29], which confers additional healthcare spending and is often an adverse prognostic indicator [30,31].

Cystic fibrosis (CF) is a complex systemic disease arising from autosomal recessive inheritance of dysregulation of the CF transmembrane conductance regulator (CFTR) membrane protein [32]. *P. aeruginosa* is the most common cause of chronic infection in people with CF, with up to 80% of individuals infected by adulthood [33]. The tremendous adaptive ability of *P. aeruginosa* is demonstrated in the intricate host–pathogen interactions which occur in acute and chronic pulmonary CF infections [34]. During acute infection, an array of virulence factors and antibiotic resistance mechanisms facilitate initial survival. An abnormally thick mucus layer and variable oxygen and nutrient availability in the CF lung [32], alongside the presence of high concentrations of antibiotics and a strong and persistent immune response, provides selective pressures and induces genetic diversification over time [35]. Further adaptation and a switch to more persistent sessile phenotypes of *P. aeruginosa* creates immotile, slow-growing colonies with a propensity to form mucoid biofilms which lead to immune evasion, colonisation, and chronic infection [36]. Therefore, despite the hostile lung environment in CF, *P. aeruginosa* can thrive to cause acute and chronic infections [6], giving rise to significant morbidity and increasing the complexity of clinical management [37]. A review by Rossi et al. further details the evolution of *P. aeruginosa* in the CF lung [34]. Additionally, innate and adaptive immune responses are greatly impacted in those with CF, allowing the bacteria to more easily cause infection while providing greater challenge for the development of novel vaccinations [38,39]. *P. aeruginosa* is a burden in other chronic respiratory conditions as well, such as severe chronic obstructive pulmonary disease and non-CF bronchiectasis and is associated with poor local immune responses and clinical outcomes [40,41].

## 4. A History of Clinical Vaccine Development against *P. aeruginosa*

Many factors have greatly hindered the development of a successful vaccine against *P. aeruginosa*. Its wealth of virulence factors, a large genome facilitating adaptation to new environments, changing phenotypes between acute and chronic infection, and the intricacies of the host immune response, among others, make this pathogen an especially challenging candidate for vaccine development. These obstacles have not dissuaded research into a vaccine; however, despite over 50 years of research, clinical vaccine development for *P. aeruginosa* has been largely unsuccessful. Along this journey, a small number of candidate vaccines using well known virulence factors as vaccine antigens have progressed to clinical trials, although none have been licensed for commercial use. Currently, there are no ongoing clinical trials investigating vaccines against *P. aeruginosa*. Table 2 outlines those studies which have progressed into phase I, II, and III trials, which will be further discussed below.

### 4.1. Lipopolysaccharide (LPS)

Alexander and Fisher published the first paper investigating *P. aeruginosa* vaccinations in a clinical setting in 1970 [42] using a heptavalent formulation of LPS extracts from seven separate serotypes of *P. aeruginosa* (Pseudogen^®^) to immunise burns patients. It was well tolerated, prevented the development of sepsis, and reduced mortality, and therefore underwent further evaluation in individuals with CF and acute leukaemia [43,44,45]. While showing promising immunogenicity, no clinical benefit was observed alongside high levels of toxicity, particularly in those with acute leukaemia. The PEV-01 vaccine used a formulation of LPS extracts from sixteen *P. aeruginosa* serotypes [46]. Studies in healthy volunteers and a large population of burn patients (*n* = 746) demonstrated that the vaccine was safe and well-tolerated and reduced mortality in both adult and paediatric burns patients [46,47]. Unfortunately, a phase II study in individuals with CF over three years did not show any reduction in colonisation [48], and individuals who were both colonised and vaccinated suffered significant and rapid clinical deterioration.

The most successful LPS-based vaccine was a conjugate vaccine consisting of a component of LPS, the O-polysaccharide from eight *P. aeruginosa* serotypes and exotoxin A named Aerugen^®^. A phase I study in healthy volunteers demonstrated safety, while the vaccine induced production of anti-LPS and anti-exotoxin IgG antibodies which functioned to both opsonize and neutralise *P. aeruginosa* and was boosted at 15 months [49,50]. A phase II study was carried out in CF patients not colonised with *P. aeruginosa* and initially showed the production of high-affinity anti-LPS and anti-exotoxin IgG, but had no impact on clinical outcome [51]. Twenty-six of the initial thirty patients were followed-up over ten years while receiving yearly booster immunisations [67]. This study showed very encouraging results, indicating a longer time to infection in those vaccinated, and overall 32% of vaccinated individuals suffered from chronic infection compared with 72% in the matched placebo control group. Vaccination was associated with better preservation of lung function and an increase in body weight [67], possibly indicating an improvement in overall health status. Previous studies using this vaccine focussed on investigating the humoral immune response. A small study in fifteen healthy volunteers found vaccination induced high levels of IFN-γ and TNF-α production from antigen-stimulated lymphocytes, indicating a more Th1-biased response [68].A large-scale randomised placebo-controlled phase III trial in CF patients was commenced; however, the study was stopped before it reached its endpoints as interim analysis revealed no significant difference between placebo and control groups [52].

### 4.2. Alginate

Individuals with CF who are not colonised with mucoid *P. aeruginosa* appear to produce opsonising and phagocytic antibodies against alginate (mucoid exopolysaccharide, MEP) [69,70]. While beneficial, these antibodies are often insufficient to eliminate the bacteria upon exposure, and prevent the establishment of future chronic infection [69]. Therefore, an enhancement of the immune response against alginate makes it an attractive candidate for therapeutic and prophylactic vaccination. In a phase I study assessing a range of MEP extracts from a mucoid strain of *P. aeruginosa*, extracts of larger molecular weight induced opsonising antibodies for up to two years post-vaccination; however, the overall immunogenicity across the study was poor [56].

### 4.3. Flagellar Antigens

Phase I and II studies using flagellar components demonstrated a significant, long-lasting increase in IgA and IgG anti-flagella antibodies in both the serum and respiratory mucosa, with a good safety profile [53,54]. A phase III double-blinded randomised placebo-controlled trial was subsequently carried out in 483 CF patients in 2007 [55] using a bivalent flagella vaccine formulation. Thirty-four percent of participants were protected against acute infection and 51% were protected against chronic infection, which unfortunately did not achieve the primary outcome of 66%. While protecting against strains with flagellar subtypes included within the vaccine formulation, this immunisation did not protect against *P. aeruginosa* serotypes with different flagellar subtypes, with 24% of the total non-vaccine flagellar subtypes identified being isolated from subjects in the intervention group [55]. Therefore, this vaccination has not undergone further development [52]. Heterogeneity in flagellar subtypes and lack of conservation across *P. aeruginosa* strains is a tremendous challenge that must be overcome to facilitate the use of flagellar antigens as vaccine candidates.

### 4.4. Whole-Cell Killed

Whole-cell killed *P. aeruginosa* vaccines have not been extensively investigated in clinical studies. An orally administered formaldehyde-inactivated strain of *P. aeruginosa* named Pseudostat^®^ is currently the only formulation which has reached clinical trials. A phase I trial in nine bronchiectasis patients demonstrated a protective immune response within the lung with the stimulation of specific lymphocytes [57]. A further phase I study in thirty healthy volunteers induced specific IgA and a sustained rise in IgG antibodies against LPS, which functioned to opsonise *P. aeruginosa* and promote phagocytosis in a human macrophage cell line [58]. Twenty subjects unfortunately experienced a range of adverse events, including upper respiratory tract infections, gastrointestinal, neurological and musculoskeletal disorders; however, they were not considered to be clinically significant or attributable to the study vaccine.

### 4.5. Outer Membrane Proteins

Outer membrane proteins F (OprF) and I (OprI) are highly immunogenic and highly conserved across all *P. aeruginosa* strains [71]. IC43 is a recombinant vaccine comprising histidine-tagged fusion proteins of conserved epitopes of OprF and OprI: Met-Ala-(His)_6_OprF_190–342_-OprI_21–83_ [61]. Preclinical investigation of IC43 in mice demonstrated a strong safety profile and was highly protective, inducing antibodies which promoted the complement-dependent opsonisation of *P. aeruginosa* [72,73]. A small phase I study in 32 healthy volunteers assessed a range of dosages from 20–500 µg adsorbed onto Al(OH)_3_ showed that the higher doses of 100 and 500 µg induced increases in vaccine-specific complement-binding and opsonophagocytic antibodies which were still present at six months post-vaccination [59]. These findings were reinforced in a small phase I study of eight burn patients which confirmed the potential of IC43 to be used in the management of burn wounds; none of the eight patients exhibited *P. aeruginosa* infection before, during, or after vaccination [60].

Further to this, a phase II trial of 401 mechanically ventilated ICU patients revealed that IC43 was most safe and tolerable at 100 µg without Al (OH)_3_ adjuvant, administered intramuscularly twice in a seven-day period [62]. This dose and formulation of IC43 resulted in the highest levels of specific vaccine-induced anti-OprF/I IgG antibodies, which remained detectable at 90 days [62]. The results of this study led to further evaluation of this vaccine formulation in a large-scale multicentre randomised placebo-controlled phase III trial in 799 medically ill mechanically ventilated ICU patients [63]. This trial assessed the safety, immunogenicity, and overall effect on invasive *P. aeruginosa* infections and mortality. Survival, all-cause mortality, and the rates of invasive or respiratory tract *P. aeruginosa* infections did not differ between the IC43 and placebo group. While achieving good immunogenicity and safety, it unfortunately showed that the efficacy of IC43 as a vaccine candidate is not evident in a large-scale study despite promising results in earlier phase trials. *P. aeruginosa* airway colonisation or infection often occurs early during an ICU stay [74], and while demonstrating high immunogenicity, IC43 only induced an IgG response following this initial colonisation. This may open a door to evaluating the role of prophylactic administration of IC43 in a high-risk population who may require frequent ICU care or admission following elective surgery, which has not yet been investigated.

A mucosal formulation of IC43 has been developed [65,66], and a phase I/II comparative study between systemic and mucosal vaccines was carried out [66]. A total of twelve healthy participants received two doses of an intranasal vaccination followed by a subsequent booster of either an intranasal or intramuscular dose. All subjects demonstrated an IgG and IgA-mediated immune response both in the serum and lower airways and the mucosal immune response was significantly higher in the intranasal booster group [66]. Similar results were obtained when a further trial was conducted in twelve individuals with chronic lung diseases [75]. Assessment of different vaccination protocols or even further assessment of the mucosal IC43 formulation has yet to be investigated in any studies outside this small phase I/II study.

## 5. New Developments in *P. aeruginosa* Vaccinology

### 5.1. Immune Response against P. aeruginosa and Its Importance in Vaccine Design

All branches of the immune system have been implicated to work in concert to protect the host against *P. aeruginosa* [76], including innate, adaptive, and mucosal responses, as shown on Figure 3. Systemic antibody and CD4^+^ T cell responses alongside local mucosal responses have all been demonstrated to play key roles in complete protection against infection [10,77,78,79].

The immune system recognises specific *P. aeruginosa* pathogen-associated molecular patterns (PAMPs) via Toll-like receptors (TLRs), which stimulate a strong initial inflammatory immune response. Lipopeptides, LPS, flagellin, and non-methylated bacterial CpG DNA are recognised by TLR2, 4, 5, and 9, respectively [80]. The initial inflammatory response recruits neutrophils and macrophages to the site of infection. The delicate balance between cellular infiltration and inflammation during the innate immune response is vital [81]. Overactive responses cause excessive host tissue damage while an underactive response results in insufficient clearance and an increased likelihood of the establishment of chronic infection [81]. The complement system plays a role in the innate response to *P. aeruginosa* as well, most notably in biofilm infections [82].

Important T cell responses have been well characterised, mainly Th1, Th2, and Th17, however these are largely underutilised in vaccine design, especially in clinical studies. B cells and their role in protection against *P. aeruginosa* are incompletely understood. In fact, high anti-pseudomonas antibody titres in patients with CF correlate with poorer clinical outcomes, and are often an adverse prognostic indicator [83].

In the field of bacterial vaccine development, in particular with *P. aeruginosa* vaccines, it remains unclear what constitutes a protective immune response. The immune response against *P. aeruginosa* is significantly complex in the CF lung. In CF patients with chronic infection, a vaccine-induced immune response has the potential to allow the clearance of persistent and recalcitrant *P. aeruginosa* infection. However, vaccination may elicit an inappropriate and undesired side effect of increased pulmonary inflammation which serves to further enhance tissue damage beyond that already observed in the CF lung [84]. This important fact is currently an issue in *P. aeruginosa* vaccine development, and must be addressed in order to ensure safety in this key patient population.

#### Advantages of Mucosal Immunity

Mucosal surfaces are the first portal of entry to the body which *P. aeruginosa* exploits to cause infection and establish colonization. Administration of a vaccine at the mucosa to promote an increased mucosal immune response to prevent or treat infection would be superior to other methods of immunisation. Furthermore, mucosal vaccines offer a multitude of advantages over traditional systemic vaccination [85]. Major advantages include:Sites of mucosal administration are typically easily accessible and highly vascularised, allowing for rapid antigen uptake;Stimulation of all arms of the immune system—antigen-specific IgA at the mucosa, systemic IgG, and cell-mediated responses;Existence of the “common mucosal immune system” confers an immune response at mucosal sites distant from the site of vaccine delivery;No requirement for needles during administration, thus potentially increasing uptake and eliminating the risk of transmitting blood-borne diseases such as hepatitis B or HIV.

While obviously advantageous, thus far only a small number of preclinical studies have investigated vaccine-induced mucosal immunity against *P. aeruginosa*, and only a single mucosal formulation has ever been trialled in humans [66,75].

### 5.2. Lessons from the Past: Enhancing and Optimising Unsuccessful Clinical Vaccines

In the field of *P. aeruginosa* clinical vaccine development there appears to be a lack of translation of encouraging early phase trial results into meaningful significant results in large cohorts of real-world patients. Therefore, alternative approaches to optimize these unsuccessful vaccine antigens should be explored. 

Whole-cell killed vaccines have been further investigated in preclinical *P. aeruginosa* vaccine development. They are attractive candidate formulations, as pathogen-associated molecular patterns (PAMPs) act as adjuvants and the cell corpuscles function as delivery systems for a range of protective antigens [86]. Hydrogen peroxide bacterial inactivation is associated with reduced toxicity, overall increased immunogenicity, and more diverse immune response when compared with formaldehyde inactivation [87]. Novel X-ray inactivated whole-cell *P. aeruginosa* vaccine greatly boosted cell-mediated immune responses, Th1 and Th2 cytokine profiles, and protected against multiple strains of *P. aeruginosa* [88]. Live attenuated vaccines function similarly to whole-cell killed formulations; however, the risk of reversion to virulence often limits their use in humans. A killed but metabolically active attenuation method has a greater safety profile [89], and was associated with a broad humoral response alongside stimulation of Th1, Th2, and Th17 cell activation.

The IC43 vaccine demonstrated high immunogenicity in clinical studies; however, it was not associated with an improvement in clinical outcomes [63]. An approach by Jing et al. showed that oligomerisation of IC43 into a heptamer led to an improved efficacy in vivo [90]. Immunisation with the resulting heptamer reduced bacterial burden and inflammation in the lungs while inducing a strong Th2 response. The potential to improve previously trialed antigens by altering their structure, conjugating to other antigens/adjuvants, or using in other delivery systems is another avenue of research. A small study in 2010 [91] showed that administration of a further IC43 vaccine dose to patients included in the 1999 clinical trial [59] elicited high OprF/I-specific antibody levels. Furthermore, vaccinated sera inhibited IFN-γ binding to *P. aeruginosa*, which provides extra information regarding the protective response induced by IC43. 

Use of single antigens has not provided an adequate immune response when used in isolation. Monomeric recombinant subunit vaccines confer protection in vivo; however, this often does not translate into humans, where multimeric vaccines perform better. Multivalent vaccines which present combinations of numerous antigens may induce a more clinically significant response. Additionally, clinical studies into alternative routes of vaccine administration in *P. aeruginosa* have not been thoroughly investigated.

### 5.3. Novel Vaccine Formulations

There are three main constituents of vaccine formulations: an antigen, to stimulate an immune response; an adjuvant, to enhance or direct a specific immune response; and a delivery system, to ensure appropriate delivery of both the antigen and adjuvant to the correct location at the correct time [92]. Herein, we highlight recent progressions in the realm of vaccine formulations, with emphasis on those which elicit a mucosal immune response.

#### 5.3.1. Antigen Discovery

All the clinical vaccines mentioned above and in Table 2 have used well-known virulence factors as antigens in subunit vaccines. In general, subunit vaccines are safer and have a lower side effect profile compared to other vaccine types [93]. Discovery of novel conserved bacterial antigens is vital to continue investigating new immunisation approaches against *P. aeruginosa*. By using various bioinformatics screens alongside whole-genome sequencing data, researchers can use reverse vaccinology to identify desirable antigens for use in vaccines. Typically, those antigens which are abundantly expressed, highly conserved and immunogenic epitopes, and surface-exposed are most useful as vaccine candidates [94].

A small number of studies have used reverse vaccinology to identify unique antigens in *P. aeruginosa*. Rashid et al. first uncovered nine novel candidates, including three uncharacterized hypothetical proteins which may provide the basis of a subunit vaccine [95]. The presence of other components of known virulence factors, including surface components of antibiotic efflux pumps, T3SS and proteins on the bacterial cell wall, validates their approach. Using reverse vaccinology and additional bioinformatic tools, Bianconi and colleagues identified 52 total potential candidates which were conserved in *P. aeruginosa* strains of varying origin and in clinical isolates from CF patients [96]. The top ten antigens were used in vivo in a murine model of acute pneumonia in 22 separate combinations. The study concluded that combinations of candidate antigens provided the best protection. This strategy of using multiple antigens has been used in previous clinical vaccines [50,63], but should undergo further investigation. Additional reverse vaccinology approaches have identified numerous candidate antigens which are yet to undergo in vivo assessment [97,98].

A Th17-based reverse vaccinology study using a select library of outer membrane and secreted proteins identified PopB, a highly conserved component of the T3SS [99]. PopB elicited Th17 responses and IL-17 production and increased bacterial clearance in lethal *P. aeruginosa* murine pneumonia in an antibody-independent manner.

To further enhance antigen discovery, methods to identify novel antigens stimulating mechanisms of cellular immunity ought to be tried. Furthermore, it must be noted that subunit vaccines in general appear to have an overall lower immunogenicity [86] and are overall more susceptible to enzymatic degradation and mucociliary clearance at the mucosal surfaces [100]. Therefore, additional measures, such as the selection of appropriate adjuvants or the use of specific delivery systems, must be considered.

#### 5.3.2. Adjuvant Selection

Adjuvants shape and enhance an antigen-specific response and are a key component of many modern vaccines. Currently, knowledge surrounding adjuvants in *P. aeruginosa* vaccines is limited due to their lack of use or complete omission in clinical studies. The most widely used adjuvant in preclinical and clinical studies is aluminium hydroxide, Al(OH)_3_, (known as alum). Phase I and II studies into IC43 concluded that seroconversion and immunogenicity was in fact highest in those patients receiving the vaccine without alum [61,62]. Subsequently, the vaccine was administered without an adjuvant in the following phase II/III trial.

Isolated use of systemic adjuvants such as alum may not be beneficial in enhancing the immune response to *P. aeruginosa*. In general, systemic adjuvants often function poorly to enhance the mucosal immune response [92]. Adjuvants such as the Th17-stimulating curdlan [99] dmLT, which potentiates antigen-specific Th1 and Th17 responses [101], or combinations of adjuvants alongside alum have provided promising preclinical results. They have not been used in clinical *P. aeruginosa* vaccine development, although many are licensed for other use in humans, as outlined by Sainz-Mejías et al. [76]. Self-adjuvanting formulations, wherein antigens are bound to an adjuvant compound, may elicit a stronger immune response. A novel formulation using dmLT [102] elicited a strong IgG and IgA response along with stimulating IL-17 production. 

#### 5.3.3. Vaccine Delivery Systems

Liposomal delivery systems have been used to administer anti-pseudomonal antibiotics in CF patients [103,104]. Their use as a vaccine delivery system demonstrated that intranasal administration increased production of pulmonary IgA in mice [105]. Liposomes have additionally been used as bacteria-free expression systems to easily and efficiently produce OprF for vaccine purposes [106], however, these liposomes are yet to be used in vivo.

Outer membrane vesicles (OMVs) are secreted by all Gram-negative bacteria, including *P. aeruginosa*. They consist of a portion of the outer membrane and other protein components which elicit a strong inflammatory response in host tissues [107]. Delivery of vaccine antigens via OMVs has been shown to stimulate potent humoral and Th1/Th17 responses in vivo [108,109], more so than isolated administration of vaccine antigens. The licensed Bexsero Meningococcal Group B vaccine uses OMV technology [110]; however, it has yet to be tested as a platform for *P. aeruginosa* antigen delivery.

Attenuated strains of *Salmonella* species have been used in a small number of studies as a delivery system for *P. aeruginosa* vaccine antigens. One clinical study in healthy volunteers found oral or intranasal administration of live attenuated *Salmonella* expressing OprF-OprI antigens led to a significant rise in specific IgA and IgG in the lower airways [111]. Consequently, further preclinical studies in mice using OprF-OprI [112] and one study using components of the T3SS [113] enhanced survival, reduced bacterial growth, and led to overall lower levels of pro-inflammatory cytokines.

Bacterial ghosts are another novel method of vaccine delivery. They are generated from Gram-negative bacteria by inducing release of their intracellular contents, resulting in an empty preserved bacterial envelope with intact surface antigens and PAMPs [114]. In the first study of its kind, Sheweita et al. developed chemically-induced BGs to vaccinate diabetic rats against *P. aeruginosa* [115]. Orally administered BGs induced a specific humoral and cell-mediated immune response. Subsequent oral challenge with P. aeruginosa demonstrated high levels of protection against systemic spread in vaccinated rats. Furthermore, a rat diabetic ulcer model showed a higher healing rate and overall complete survival when comparing vaccinated and unvaccinated groups. Interestingly, generation of BGs permits conservation of key Gram-negative external bacterial components. This can potentially induce cross-protectivity against other Gram-negative bacteria.

#### 5.3.4. Nanoparticles

Study of various biopolymer and nanoparticle delivery systems shows promise to enhance subunit antigen immunogenicity for potential use in humans. Nanoparticles have undergone extensive research for antigen delivery through a range of routes, including oral, parenteral, and more recently, topical. Dissolving microneedle patches were used as a platform to deliver heat-inactivated *P. aeruginosa* [116]. This study showed encouraging properties and significantly reduced bacterial burden in the lungs of vaccinated mice when compared with unvaccinated controls. While only a proof of concept, this study demonstrates the potential use of nanoparticles as components of unique delivery systems for easy and cost-effective approaches to vaccination.

Mannose-modified chitosan microspheres delivering OprF-OprI antigens elicited specific systemic and mucosal humoral responses [117]. Furthermore, intranasal immunisation established potent Th1/Th2 cellular immunity. Gold nanoparticles enhanced the specific antibody production and the immunogenicity of an exotoxin A and detoxified LPS conjugate vaccine [118]. Aluminium (oxy) hydroxide nanorods delivering whole-cell killed PAO1 promoted a rapid initial immune response [119], which may prove beneficial in preventing colonisation in the lung. Evaluation of two unique nanoparticle formulations using PopB [120] demonstrated that addition of a novel TLR4 agonist, BECC438 (a detoxified lipid A analogue), provided the highest protective efficacy in mice, offering an initial step into optimising this formulation for use in humans. Polylactic co-glycolic acid (PLGA) is a promising vaccine delivery system for the encapsulation of *P. aeruginosa* antigens. Conjugation with exotoxin A resulted in a stable and highly immunogenic vaccine. This resulted in a potent cellular and humoral immune response, and conferred greater protection and cytokine and antibody production compared to exotoxin A alone [121]. Similarly, a PLGA vaccine using alginate antigen showed significantly increased opsonizing antibody titres and reduced bacterial burden compared to vaccination with only alginate [122].

Biopolymers adsorbed with various combinations of ten vaccine candidate antigens all conferred protection in a murine model of acute pneumonia [123]. Moreover, the best performing antigen combination induced a 30% increase in survival (up to 90%) when administered intranasally compared to intramuscular administration. Intranasal vaccine delivery both stimulates a stronger local mucosal immune response and is a much easier approach for vaccine administration and distribution.

## 6. Conclusions

Preclinical and clinical studies have demonstrated the extremely challenging nature of developing a vaccine against *P. aeruginosa*. Most vaccines to date have used traditional methods of vaccine design involving the use of well-studied virulence factors such as vaccine antigens. Further development of many of the vaccines outlined above has likely ended, and new methods of vaccine candidate discovery, namely, reverse vaccinology, should be employed. Complex and virulent *P. aeruginosa* infections cause high morbidity and mortality in many populations, and the emergence of MDR strains will continue to be a challenge for patients and health professionals worldwide. Development of a vaccine against this pathogen will help to prevent infections and ultimately save lives for years into the future.

## Figures and Tables

**Figure 1 vaccines-10-01100-f001:**
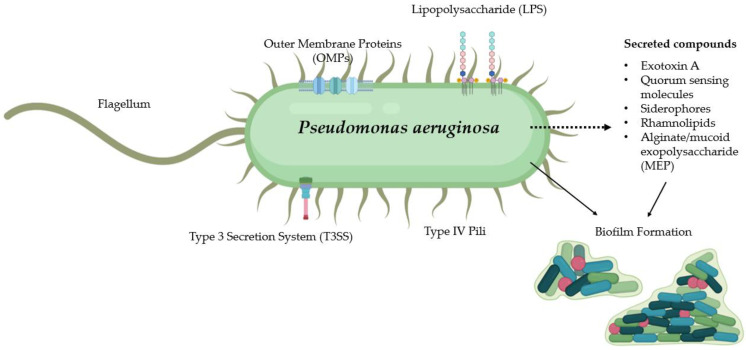
Key virulence factors important in the pathogenesis of *P. aeruginosa* infections.

**Figure 2 vaccines-10-01100-f002:**
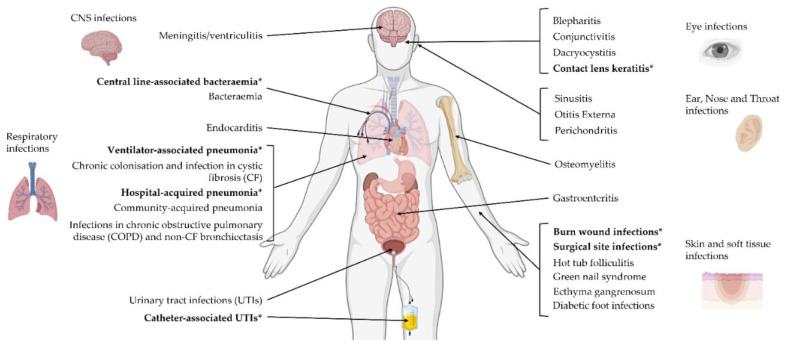
The many sites of *P. aeruginosa* infection throughout the body. Labels in bold and starred (*) are those infections associated with healthcare or healthcare devices.

**Figure 3 vaccines-10-01100-f003:**
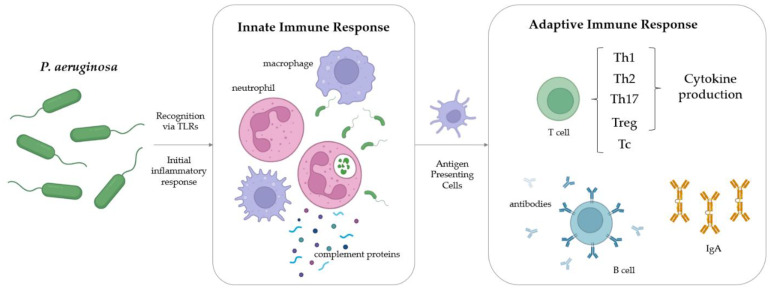
A summary of the key aspects of the host immune response against *P. aeruginosa* infection.

**Table 1 vaccines-10-01100-t001:** Mechanisms of antimicrobial resistance in *P. aeruginosa.*

Intrinsic	Acquired	Adaptive
Restricted outer membrane permeability	Mutational changes	Biofilm formation
Antibiotic-inactivating enzymes	Over-expression of resistance genes	Persister cells
Efflux pumps	Horizontal transfer of resistance genes	

**Table 2 vaccines-10-01100-t002:** A summary of all clinical vaccine trials in *P. aeruginosa.*

Antigen	Formulation	Phase	Dosage, Administration, Adjuvant	Population	Outcomes	Study and Reference
**Lipopolysaccharide**	LPS extracts from 7 PA serotypes (Pseudogen^®^)	II	NS	72 burns	Prevented development of sepsis and subsequent death	Alexander and Fisher (1970) [42]
		II	NS, IM, 5 doses, none	361 cancer	Reduced mortality, slight reduction in fatal PA infection but associated with adverse events	Young et al. (1973) [43]
		II	6–12 µg/kg, IM, 6 doses, none	12 CF, 22 acute leukaemia	No clinical benefit, CF patients showed antibody response adverse events in 95% leukaemia patients	Pennington et al. (1975) [44,45]
	LPS extracts from 16 PA serotypes (PEV-01)	I	0.5 mL, SC, 3 doses, none	15 healthy	Variable antibody response, no toxic adverse events	Jones et al. (1976) [46]
		II	0.5 mL, SC, 3 doses, none	746 burns	Reduced mortality in adults and children, variable antibody response, overall increase in bactericidal capacity of blood	Roe and Jones (1983) [47]
		II	0.25, 0.5 mL, SC, 3 doses, none	34 CF	No reduction in colonization, vaccinated and colonized individuals suffered most rapid deterioration	Langford and Hiller (1984) [48]
	LPS extracts from 8 PA serotypes conjugated to Exotoxin A (Aerugen^®^)	I	0.5 mL, SC, 2 doses, none	20 healthy	Safe, anti-exotoxin and anti-LPS IgG produced, boosted at 15 months	Cryz et al. (1987, 1988) [49,50]
		II	6–12 µg/kg, IM, 3 doses, none	30 CF, non-colonised	High affinity IgG response to exotoxin and LPS, no change in clinical status	Schaad et al. (1991) [51]
		III	NS	476 CF	No clinical difference between groups at interim analysis—study stopped	Döring (2010) [52]
**Flagellum**	Monovalent	I	40 µg, IM, NS, Al(OH)_3_	220 healthy	High serum and respiratory mucosal anti-flagella antibody titres	Crowe et al. (1991) [53]
		II	40 µg, IM, NS, none	10 healthy		Döring et al. (1995) [54]
	Bivalent	III	40 µg, IM, 4 doses, Al(OH)_3_ and thiomersal	483 CF	Long-lasting serum anti-flagella serotype-specific antibodies, 34% protected against acute infection, 51% protected against chronic infection	Döring et al. (2007) [55]
**Alginate**	2 preparations of MEP extracts from mucoid PA	I	10, 50, 100, 150 µg, IM, 2 doses, none	28 healthy	Poorly functioning opsonizing antibodies, not augmented by booster dose	Pier et al. (1994) [56]
**Whole-cell killed**	Pseudostat^®^	I	NS, PO, NS, NS	9 bronchiectasis	Induction of specific lymphocyte response, decrease in bacterial sputum counts	Cripps et al. (1997) [57]
		I	150 mg, PO, 2 doses, none	30 healthy	IgG and IgA opsonizing antibodies, 20 adverse events	Cripps et al. (2006) [58]
**Outer Membrane Proteins**	OprF-OprI systemic formulation (IC43)	I	20, 50, 100, 500 µg, IM, 4 doses, Al(OH)_3_	32 healthy	Complement binding and opsonizing antibodies present at 6-months post 3rd vaccine	McGhee et al. (1999) [59]
		I	100 µg, IM, 3 doses, Al(OH)_3_	8 burns	Well-tolerated, seroconversion in 7 subjects	Mansouri et al. (2003) [60]
		I	50, 100, 200 µg, IM, 2 doses, Al(OH)_3_	163 healthy	Safe and well-tolerated, induced specific IgG response vs placebo, higher doses were not more effective	Westritschnig et al. (2014) [61]
		II	100, 200 µg, IM, 2 doses, Al(OH)_3_	401 mechanically ventilated ICU	Increased IgG persisting until day 70 post-vaccination, not powered to assess infection rates/mortality	Rello et al. (2017) [62]
		II/III	100 µg, IM, 2 doses, none	799 mechanically ventilated ICU	Well-tolerated and immunogenic, no difference in survival or mortality vs. placebo	Adlbrecht et al. (2020) [63]
	OprF-OprI mucosal formulation (IC43)	I	500 µg, IN, 3 doses	8 healthy	Safe and well-tolerated, 6 subjects showed increased serum IgG and IgA	Larbig et al. (2001) [64]
	OprF-OprI systemic and mucosal formulations (Comparative study)	I/II	Mucosal 1 mg, IN, 2 doses (+1 booster), noneSystemic 100 µg, IM, (1 booster), Al(OH)_3_	12 healthy (6 mucosal only, 6 mucosal with systemic booster)	Safe and well-tolerated, immunogenic in all, serum IgG higher with systemic booster	Göcke et al. (2003) [65] Baumann et al. (2007) [66]

## Data Availability

Not applicable.

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
