# Peer review of "Pseudomonas aeruginosa: Recent Advances in Vaccine Development"

_vaccines, 2022, doi:10.3390/vaccines10071100_

Round 1

Reviewer 1 Report

This is a descriptive review, but on a topic of great relevance for several decades, given the high prevalence and severity of P. aeruginosa infections, mainly affecting immunocompromised patients. The manuscript is clear and well written. However, I believe it would need a revision of the text, as there is a need for “commas” in several sentences. Regarding its organization, the authors made a very adequate division of the items, including a brief introduction about the microorganism, its virulence factors and the infections it causes. Then the authors addressed the historical aspects of vaccines against P. aeruginosa since the first study in 1970 and described other vaccine candidates made with different bacterial components or products and concluded the article mentioning the new developments regarding vaccines against this pathogen. Therefore, I am in favor of its publication and suggest that the manuscript undergoes a revision with respect to a more adequate punctuation.

Author Response

Reviewer 1

This is a descriptive review, but on a topic of great relevance for several decades, given the high prevalence and severity of P. aeruginosa infections, mainly affecting immunocompromised patients. The manuscript is clear and well written. However, I believe it would need a revision of the text, as there is a need for “commas” in several sentences. Regarding its organization, the authors made a very adequate division of the items, including a brief introduction about the microorganism, its virulence factors and the infections it causes. Then the authors addressed the historical aspects of vaccines against P. aeruginosa since the first study in 1970 and described other vaccine candidates made with different bacterial components or products and concluded the article mentioning the new developments regarding vaccines against this pathogen. Therefore, I am in favor of its publication and suggest that the manuscript undergoes a revision with respect to a more adequate punctuation.

Response: We thank the reviewer for their comments and for highlighting the requirement for more adequate punctuation and have addressed this throughout.

Reviewer 2 Report

Comments and Suggestions for Authors

In this review, (Vaccines-1761735) entitled " Pseudomonas aeruginosa: Recent Advances in Vaccine Development", Killough et al., described the advances in vaccine development of Pseudomonas aeruginosa. This article is well written and discussed the undermined aspects of Pseudomonas aeruginosa vaccine development. However, authors need to address the following concerns before publication.

1.      Pseudomonas aeruginosa was identified as a common coinfecting pathogen in COVID-19 patients causing exacerbation of illness. Please talk about this issue in Introduction section.

  1. The authors haven’t stated the potential clinical side effects of Pseudomonas aeruginosa vaccines (if any) post-administration in vivo and in vitro It is encouraged to mention human clinical trial information pertaining vaccine in current ongoing or FDA-approved therapies.
  2. The cell and tissue uptake percentage and assimilations of various BGs need to be discussed.
  3. The cellular inflammatory response because of vaccine Pseudomonas aeruginosa uptake was not included. Specifically, the cytokine induction and other response factors need to be discussed.
  4. An interesting figure should be added to illustrate the immune response.

6.      The authors haven’t stated the Pseudomonas aeruginosa Ghosts vaccine as one of the most important and recent topics.

  1. Although the authors talk about Nanoparticle vaccine for Pseudomonas aeruginosa, the subject need more details under a separate title

Author Response

Reviewer 2

In this review, (Vaccines-1761735) entitled " Pseudomonas aeruginosa: Recent Advances in Vaccine Development", Killough et al., described the advances in vaccine development of Pseudomonas aeruginosa. This article is well written and discussed the undermined aspects of Pseudomonas aeruginosa vaccine development. However, authors need to address the following concerns before publication.

1. Pseudomonas aeruginosawas identified as a common coinfecting pathogen in COVID-19 patients causing exacerbation of illness. Please talk about this issue in Introduction section.

Response: We thank reviewer for highlighting the importance of including this important information. We have added this to the introduction section, as requested.

2. The authors haven’t stated the potential clinical side effects of Pseudomonas aeruginosa vaccines (if any) post-administration in vivo and in vitro It is encouraged to mention human clinical trial information pertaining vaccine in current ongoing or FDA-approved therapies.

Response: This is an important consideration and we have now included a discussion regarding the potential for increased inflammatory response in chronic infection. There are currently no ongoing clinical trials and we have included a line reflecting this at the beginning of the vaccine development section.

3. The cell and tissue uptake percentage and assimilations of various BGs need to be discussed.

Response: We thank the reviewer for their comment, but we don’t have information on this with regards to pseudomonas. Perhaps clarification can be provided if of relevance.

4. The cellular inflammatory response because of vaccine Pseudomonas aeruginosa uptake was not included. Specifically, the cytokine induction and other response factors need to be discussed.

Response: We thank the reviewer for highlighting this and have subsequently included information regarding this within the text.

5. An interesting figure should be added to illustrate the immune response.

Response: We thank review for this great suggestion, which adds to the review article, we have subsequently included a figure illustrating the immune response.

6. The authors haven’t stated the Pseudomonas aeruginosa Ghosts vaccine as one of the most important and recent topics.

Response: We have subsequently included an additional section highlighting Pseudomonas aeruginosa ghost vaccine

7. Although the authors talk about Nanoparticle vaccine for Pseudomonas aeruginosa, the subject need more details under a separate title

Response: We have added an additional section providing further more in depth discussion on nanoparticle formulations.

Reviewer 3 Report

The review paper entitled ‘Pseudomonas aeruginosa: Recent Advances in Vaccine Development’ by Killough et al. gives a concise and interesting summary of attempts to  establish vaccines against the opportunistic pathogen Pseudomonas aeruginosa. Therefore, the authors screened literature over the last 50 years and compiled a list of 22 studies which reached at least phase I of a clinical study. Although, Killough and coworkers do not claim the completeness of their overview, an own search brought some more interesting publications to light and at least some of the more recent ones might be mentioned in this review (see list attached). Nevertheless, I want to mention that my screen was not that intensive and I may have missed some links between the studies.

In their introduction, the authors try to give an overview on P. aeruginosa virulence factors. However, due to the complexitiy of the topic this has to be done in a rather limited way. The authors themselves already refer to other reviews, so, I wonder, if limiting the review to the relevant aspects aimed at the vaccination against P. aeruginosa might be the better choice. For example, biofilm and NET formation might be excluded, as these are only touched on very briefly anyway.

The review is well written, interesting and enjoyable to read. In particular, the authors’ comments on problems in vaccine development, such as an underutilization of the T-cell response, to my mind, add to the value of the review. The overview on novel vaccine formulations is concise and well-structured according to major vaccine constituents antigens, adjuvants and delivery systems. The mention of the technology of reverse vaccinology is also useful and should be revisited in the conclusions because of its importance.

In summary, I would recommend the review for publication after some minor changes, since to my mind it is nice to read for the interested layperson and a good update for people working in the field.

Minor comments:

Figure 2. Infections are not completely allocated to organ systems. Please complete (CNS, airways), P. aeruginosa also plays an important role in COPD patients, those should be mentioned here as well.

Page 1, line 34: Wording sounds too hyperbolic, I would prefer a more factual expression.

Page 2, line 66: NET formation is described but not named. Please do so.

Page 4, line 99, supernumerary blank

Page 4, line 120: It is incorrect to call CF a syndrome, since it is an inherited disease.

Page 13, line 372: spelling error ‘select library’

Format of references should be checked. Italics often is written with <i>

List of clinical trials dealing with P. aeruginosa vaccines.   

1)    Hunt JL, Purdue GF. A clinical trial of i.v. tetravalent hyperimmune Pseudomonas globulin G in burned patients. J Trauma. 1988 Feb;28(2):146-51.           

2)    Ding B, von Specht BU, Li Y. OprF/I-vaccinated sera inhibit binding of human interferon-gamma to Pseudomonas aeruginosa. Vaccine. 2010 Jun 7;28(25):4119-22.

3)    Jang IJ, Kim IS, Park WJ, Yoo KS, Yim DS, Kim HK, Shin SG, Chang WH, Lee NG, Jung SB, Ahn DH, Cho YJ, Ahn BY, Lee Y, Kim YG, Nam SW, Kim HS. Human immune response to a Pseudomonas aeruginosa outer membrane protein vaccine. Vaccine. 1999 Jan;17(2):158-68.

4)    von Specht B, Knapp B, Hungerer K, Lücking C, Schmitt A, Domdey H. Outer membrane proteins of Pseudomonas aeruginosa as vaccine candidates. J Biotechnol. 1996 Jan 26;44(1-3):145-53.

5)    Lee NG, Jung SB, Ahn BY, Kim YH, Kim JJ, Kim DK, Kim IS, Yoon SM, Nam SW, Kim HS, Park WJ. Immunization of burn-patients with a Pseudomonas aeruginosa outer membrane protein vaccine elicits antibodies with protective efficacy.  Vaccine. 2000 Mar 17;18(18):1952-61.

6)    Kim DK, Kim JJ, Kim JH, Woo YM, Kim S, Yoon DW, Choi CS, Kim I, Park WJ, Lee N, Jung SB, Ahn BY, Nam SW, Yoon SM, Choi WJ. Comparison of two immunization schedules for a Pseudomonas aeruginosa outer membrane proteins vaccine in burn patients. Vaccine. 2000 Dec 8;19(9-10):1274-83.

7)    Bbukowska D, Serafińska D, Rudowski W, Hoffman S, Olański W, Gardzińska E, Popiel D, Jedrzejczak G, Czarnecka I. [Results of using polyvalent Pseudomonas aeruginosa vaccine in children with burns by various medical centers]. Pol Tyg Lek. 1989 Oct 23-Nov 6;44(43-45):924-7.

8)    Zuercher AW, Imboden MA, Jampen S, Bosse D, Ulrich M, Chtioui H, Lauterburg BH, Lang AB. Cellular immunity in healthy volunteers treated with an octavalent conjugate Pseudomonas aeruginosa vaccine. Clin Exp Immunol. 2006 Jan;143(1):132-8.

9)    Yang HP, Wang TC, Wang SJ, Chen SP, Wu E, Lai SQ, Chang HW, Liao CW. Recombinant chimeric vaccine composed of PRRSV antigens and truncated Pseudomonas exotoxin A (PE-K13). Res Vet Sci. 2013 Oct;95(2):742-51.

Author Response

The review paper entitled ‘Pseudomonas aeruginosa: Recent Advances in Vaccine Development’ by Killough et al. gives a concise and interesting summary of attempts to  establish vaccines against the opportunistic pathogen Pseudomonas aeruginosa. Therefore, the authors screened literature over the last 50 years and compiled a list of 22 studies which reached at least phase I of a clinical study. Although, Killough and coworkers do not claim the completeness of their overview, an own search brought some more interesting publications to light and at least some of the more recent ones might be mentioned in this review (see list attached). Nevertheless, I want to mention that my screen was not that intensive and I may have missed some links between the studies.

Response: We thank reviewers for highlighting this. We have subsequently included further information regarding additional studies within the text.

In their introduction, the authors try to give an overview on P. aeruginosa virulence factors. However, due to the complexitiy of the topic this has to be done in a rather limited way. The authors themselves already refer to other reviews, so, I wonder, if limiting the review to the relevant aspects aimed at the vaccination against P. aeruginosa might be the better choice. For example, biofilm and NET formation might be excluded, as these are only touched on very briefly anyway.

The review is well written, interesting and enjoyable to read. In particular, the authors’ comments on problems in vaccine development, such as an underutilization of the T-cell response, to my mind, add to the value of the review. The overview on novel vaccine formulations is concise and well-structured according to major vaccine constituents antigens, adjuvants and delivery systems. The mention of the technology of reverse vaccinology is also useful and should be revisited in the conclusions because of its importance.

Response: This is an important point to highlight and we thank reviewers for doing so, we have subsequently included reverse vaccinology in the conclusion

In summary, I would recommend the review for publication after some minor changes, since to my mind it is nice to read for the interested layperson and a good update for people working in the field.

Minor comments:

Figure 2. Infections are not completely allocated to organ systems. Please complete (CNS, airways), P. aeruginosa also plays an important role in COPD patients, those should be mentioned here as well.

Page 1, line 34: Wording sounds too hyperbolic, I would prefer a more factual expression.

Page 2, line 66: NET formation is described but not named. Please do so.

Page 4, line 99, supernumerary blank

Page 4, line 120: It is incorrect to call CF a syndrome, since it is an inherited disease.

Page 13, line 372: spelling error ‘select library’

Format of references should be checked. Italics often is written with <i>

Response: We thank reviewers for highlighting all of the above, we have addressed all minor comments accordingly.  

Round 2

Reviewer 2 Report

The review paper accepted in present form